# Edible Plant Sprouts: Health Benefits, Trends, and Opportunities for Novel Exploration

**DOI:** 10.3390/nu13082882

**Published:** 2021-08-21

**Authors:** Simon Okomo Aloo, Fred Kwame Ofosu, Sheila M. Kilonzi, Umair Shabbir, Deog Hwan Oh

**Affiliations:** 1Department of Food Science and Biotechnology, College of Agriculture and Life Sciences, Kangwon National University, Gangwon-do, Chuncheon 24341, Korea; okomosimon@gmail.com (S.O.A.); fkofosu17@gmail.com (F.K.O.); umair336@gmail.com (U.S.); 2Department of Food Science and Nutrition, School of Agriculture and Biotechnology, Karatina University, Karatina 10101, Kenya; skilonzi@gmail.com

**Keywords:** germination, sprouts, alfalfa, buckwheat, broccoli, red cabbage, metabolites, technology

## Abstract

The consumption of plant sprouts as part of human day-to-day diets is gradually increasing, and their health benefit is attracting interest across multiple disciplines. The purpose of this review was to (a) critically evaluate the phytochemicals in selected sprouts (alfalfa, buckwheat, broccoli, and red cabbage), (b) describe the health benefits of sprouts, (c) assess the recent advances in sprout production, (d) rigorously evaluate their safety, and (e) suggest directions that merit special consideration for further novel research on sprouts. Young shoots are characterized by high levels of health-benefitting phytochemicals. Their utility as functional ingredients have been extensively described. Tremendous advances in the production and safety of sprouts have been made over the recent past and numerous reports have appeared in mainstream scientific journals describing their nutritional and medicinal properties. However, subjects such as application of sprouted seed flours in processed products, utilizing sprouts as leads in the synthesis of nanoparticles, and assessing the dynamics of a relationship between sprouts and gut health require special attention for future clinical exploration. Sprouting is an effective strategy allowing manipulation of phytochemicals in seeds to improve their health benefits.

## 1. Introduction

As people become increasingly conscious about the relationship between diets and health, attention is shifting towards assessing better methods to improve the functionality of foods. In the recent past, there has been a growing popularity of sprouted edible seeds in human diets [1]. Today, there is an increased accumulation of a vast store of knowledge relating to the therapeutic properties of sprouted foods; what is more, with the recent coronavirus outbreak, the demand for functional foods to improve body immunity is on the rise [2]. Initially, germinated legume seeds were the major type of sprouts consumed in the human diet whereas sprouted cereal grains have been mainly utilized as fodder for animals [3]. However, currently, a diverse range of sprouted foods originating from a broad range of seeds such as alfalfa, buckwheat, red cabbage, and broccoli sprouts have become popular and are widely consumed across the globe [4,5]. The growing popularity observed for sprouts is mainly due to their positive health impact. Sprouts have been associated with a variety of biologically active constituents with potential health benefits. For instance, Shi et al. stated that the increasing consumption of alfalfa sprouts is due to their high content of saponins and other useful bioactive compounds present in the germinated seeds [6]. Such compounds are known to possess antioxidant activity, antiviral activity, immune stimulant activity, and antidiabetic activity, among other functions in both humans and animals [6]. Across Asia, mostly in countries such as Japan, China, and Korea, the consumption of buckwheat sprouts in the form of noodles is significantly increasing [4]. Sprouted buckwheat is well-known for its antioxidant, antihypocholesterolemic, and neuroprotective functions [7] while red cabbage and broccoli sprouts are popular brassica vegetables that exhibit antimicrobial, anticancer, as well as anti-obesity properties [5].

Over decades, significant progress has been made concerning sprout production. More sophisticated technologies have been assessed, and examinations of various methods to enhance the functional properties of edible plant sprouts are still underway [1]. Likewise, the scientific literature on the production and health benefits of germinated edible seeds has dramatically increased. Reviews on new developments on sprouts have been published [7,8,9]. Nevertheless, despite this vast knowledge about the relationship between sprouts and health, there are still aspects of sprouts that require more emphasis. A more updated assessment of active metabolites, recent advances in technologies, and new applications of germinated seeds is still required. The assessment is especially beneficial to sprout developers who are in constant need of updated information to enable them to produce cheaper but high-quality seed germinates. This review offers a summary of the phytochemicals present in sprouts with a focus on legumes (alfalfa), cereals (buckwheat), and vegetable (red cabbage and broccoli) sprouts and their health benefits. The review provides implications of recent technologies in enhancing the biological activity and safety of sprouted seeds. In addition, an attempt is made to identify intriguing areas of investigation on edible sprouts that merit special consideration for further research.

## 2. Germination Process of Seeds

Germination is a common physical method that has been used over the past to enhance the safety and nutritional value of edible seeds [10]. The processing method activates enzymes in a dormant seed and triggers various enzymatic activities leading to the breakdown of stored proteins, carbohydrates, and lipids into simpler forms [11]. During the process, the degradation of sugars, free amino acids, and organic acids is significantly increased [12]. The processes ultimately increase the bioavailability of active compounds in sprouted seeds. The complex physical and metabolic changes that occur during the process of germination can be grouped into decontamination, soaking, and sprouting stages. Due to the possible contamination of seeds during the handling process, chemicals including calcium hypochlorite and sodium hypochlorite are commonly used to kill microbes on seeds before sprouting [13].

Soaking is usually performed using water or any other soaking solution such as a salt solution to allow seeds to absorb moisture and rehydrate before germination. Soaking provides a conducive environment for the growth of bacteria because bacteria grow best in the presence of moisture, hence, it is necessary to change soaking water frequently to prevent the growth of microorganisms [14]. Excessive soaking may enhance the microbial accumulation and the fermentation of seeds while insufficient soaking does not support the augmentation of phytochemical content in seeds [14]. Therefore, parameters such as seed weight/water volume ratio, time, and temperature of soaking are monitored for optimum soaking. In general, depending on the characteristics of different seeds, soaking can be performed for up to 24 h at room temperature [12]. Adding a little salt to the soaking solution and aeration during soaking can improve the absorption of water by soaked seeds [15]. The sprouting stage is key and sensitive since it determines the final component of sprouted seeds. Several factors such as light, temperature, time, and moisture must be closely monitored because they are responsible for changes in seeds during sprouting. Generally, sprouting is commonly performed in the dark and the temperature is maintained between 10–20 °C for most seed species [16]. Sprouting time depends on the purpose of the sprout. However, for most edible seeds, the time ranges between 3 and 5 days during which significant changes take place in the seed composition and mature sprouts are harvested [16].

## 3. Phytochemicals in Selected Edible Plant Sprouts

Sprouts are a potential natural source of diverse bioactive compounds with various health-benefitting effects in the prevention and treatment of diseases [17]. Microcomponents of alfalfa sprouts which include trace elements such as copper (Cu), manganese (K), and selenium (Se) play fundamental roles in controlling oxidative stress and free radical balance in various physiological processes. Mn, a constituent of manganese superoxide dismutase (Mn-SOD), is an enzyme which averts the effects of free radicals on mitochondria [18]. The high concentration of Mn in alfalfa sprouts can aid in stimulating insulin secretion and improve insulin function in diabetic patients [19]. On the other hand, Cu is a component of cytochrome oxidase and plays a critical role as a free oxygen scavenger. Se forms a structural part of several glutathione peroxidase enzymes which act as regulators in the redox state of various biomolecules [18]. Alfalfa sprouts are also rich in vitamins such as vitamin B complex [20], vitamins C and E [21]. Moreover, a diverse number of phenolic compounds (gallic and caffeic acids), flavonoids (apigenin, kaempferol, myricetin, naringin quercetin, rutin, daidzein, and genistein) can be found in a substantial amount in alfalfa sprouts [22]. These compounds are responsible for antidiabetic, anti-obesity, antioxidant, as well as many other biological activities. Other non-phenols such as the saponin component of alfalfa play key biological functions in the body [23]. Saponins and their derivatives such as prosapogenins and sapogenins have been reported to exert a high antimicrobial activity against yeasts and bacterial strains [24]. Studies also reveal that saponins can inhibit cholesterol esterase, acetyl coenzyme, and carboxylase enzymes, thereby preventing fatty acid synthesis in the body [19]. The inhibitory function of saponins on fatty acids synthesis helps balance the ratio between high-density lipoprotein (HDL) cholesterol and low-density lipoprotein (LDL) cholesterol.

Buckwheat is a cereal plant belonging to family Polygonaceae containing approximately 1200 species [25]. It exists as common buckwheat (*Fagopyrum esculentum* Moench) and Tartary buckwheat (*Fagopyrum tataricum* Gaertn.) [26]. Buckwheat is considered to be a valued source of high-quality proteins, fats, dietary fibers, as well as mineral nutrients [27]. Common buckwheat sprouts contain abundant flavonoids including orientin, vitexin, rutin, and their derivatives [28]. Anthocyanins, C-glycosylflavones (orientin, isoorientin, vitexin, and isovitexin), rutin, and quercetin are involved in the antioxidant activity of buckwheat [29,30]. Tartary buckwheat is composed of rutin as the major flavonoid component playing key roles in various health-promoting activities [31]. Other compounds including vitamins C and E, β-carotene, and γ-aminobutyric acid (GABA) have been shown to possess potential health benefits in buckwheat [32,33,34].

Vegetables of the family *Brassica* such as broccoli and red cabbage are of great interest in nutrition. Studies have linked the intake of *Brassica* vegetables to reduced health risks related to aging [7]. Among the ingredients responsible for the health effects of these vegetables are phenolic compounds such as anthocyanins [35,36]. Red cabbage is specifically rich in acylated anthocyanins responsible for the positive effects on the gastrointestinal tract [37,38]. Broccoli sprouts contain gallic, kaempferol, chlorogenic, sinapinic benzoic, quercetin, and ferulic have been shown to exert health benefits in the body [39]. The major non-phenolic compound reported in almost all the *Brassica* vegetables is glucosinolates (GLs) [40]. Glucosinolates (GLs) are synthesized from a small number of primary amino acids including tyrosine, phenylalanine, and tryptophan [41]. These metabolites are found inside vacuoles and can be degraded by the myrosinase enzyme into simpler, active forms such as isothiocyanates and thiocyanates [41,42]. Several GLs have been identified in broccoli [43]. Among these, glucoraphanin is the dominant and most known GLs in broccoli sprouts composing 81% of the total GLs content [44]. GLs in the vegetables are inactive, but they can be hydrolyzed to generate the active form, sulforaphane (4-methylsulfinylbutyl isothiocyanate) in plants and upon digestion in humans [45]. Myrosinase enzyme, a family of enzymes involved in plant defense mechanism and also present in the human gut catalyzes the conversion of GLs into active form improving their bioavailability [46].

## 4. Health Benefits of Sprouts

### 4.1. Antioxidant Activity

The antioxidant activity is determined by means of ferric reducing antioxidant power (FRAP), 1,1-diphenyl-2-picryl-hydrazyl antioxidant assay (DPPH), Trolox equivalent antioxidant capacity (TEAC), or 2,2′-azino-bis-3-ethylbenzothiazoline-6-sulfonic acid radical scavenging assay (ABTS). As a result of their therapeutic potential against radical damages, the antioxidant activity of plant extracts has been a major focus of research. Numerous phenolic and nonphenolic compounds in sprouts have been identified to have antioxidant activities. The activity of ascorbic acid in sprouts has been described [21]. Similarly, GLs in broccoli and red cabbage have been shown to exhibit radical scavenging activities [47]. Consequently, sprouts have gradually received appreciation for their functional properties.

### 4.2. Cytotoxic Activity

Human exposure to chemicals and nanoparticles is inevitable since these substances are frequently encountered in day-to-day life. Some chemicals as well as some nanoparticles can cause substantial cytotoxic effects in the body. Recently, exposure to toxic substances has led to a rise in cancer as a global health concern. In 2018 only, there were 599,274 cancer-related deaths in the United States [48]. Due to the high mortality rate of cancer patients, scientists have investigated the role of plant sprouts in cancer management. A study by Gawlik-Dziki et al. on the effect of phenolic compounds on anticancer activity of broccoli sprouts revealed a significant inhibitory activity of sprouts on the progression of prostate cancer [39]. Drozdowska et al. also demonstrated the ability of young shoots of red cabbage to exert higher anticancer activity compared to mature vegetables due to the high content of GLs in young sprouts [49]. Some reports revealed that the alteration of gene expression induced by active compounds in buckwheat is one of the contributing factors to anticancer inhibitory effects of buckwheat sprouts [7]. The phenolic compounds in buckwheat including rutin and quercetin can induce apoptosis of cancer cells, cell cycle arrest (limiting cells from growing to the G1 phase), prevent cytotoxicity, as well as inhibit migration and progression of cells [50,51]. Alfalfa L-canavanine possesses a potent inhibitory effect against cancer cells [52]. Moreover, 3-terpene derivatives and 5-flavonoid [53], β-carotene, and lutein [20] have been reported as anticancer phytochemicals in alfalfa making sprouted seeds key targets in cancer chemoprevention and therapy.

### 4.3. Antidiabetic Activity

Diabetes mellitus is a group of metabolic diseases characterized by elevated blood sugar levels (hyperglycemia). The disease originates from multiple factors involving either defects in insulin secretion or errors in insulin action, and sometimes both of these incidences may simultaneously lead to hyperglycemia [54]. The interest in finding inhibitors that can block or delay carbohydrate hydrolysis with enzymes such as alpha-glucosidases and reduce the accumulation of sugar is shaping research in diabetes treatment [55]. Studies have demonstrated that most natural antioxidants in plant sprouts can exert the action of defense mechanisms against oxidative stress and inhibit primary enzymes which hydrolyze carbohydrates into simple sugars [56,57,58,59]. In addition, the formation of advanced glycation end-products (AGEs) is prevalent in diabetic patients and can contribute to the development of osteoporosis [60]. A broccoli sprout extract was reported to possess a substantial role in preventing the formation of AGEs by inhibiting inflammatory reactions in endothelial cells [61]. In another study, 5% SFN-rich broccoli sprout extract significantly lowered the formation of AGEs in vitro [62].

### 4.4. Hypocholesterolemic and Anti-Obesity Activity

Increased cholesterol intake can induce oxidative stress in the body leading to elevation of low-density lipoproteins (LDL) and their oxidized form (oxLDL). This can subsequently lead to the development of atherosclerosis and other related cardiovascular diseases [7]. Both in vitro and in vivo studies have supported the role of sprouted seeds in protective effects against heart-related diseases caused by imbalanced cholesterol levels [63]. Lin et al. examined the hypolipidemic activity of buckwheat seeds and sprouts [63]. The study indicated that serum levels of LDL cholesterol were significantly decreased by buckwheat meals (seed and sprout meals), indicating a potent inhibitory effect of buckwheat sprouts against the hypolipidemic condition [63]. On the other hand, alfalfa sprouts have been associated with an inhibitory effect against cholesterol absorption and their reduction in the blood plasma [64,65]. The hypocholesterolemic activity of alfalfa sprouts has been related to increased conversion of hepatic cholesterol to bile salts by alfalfa saponins, leading to their loss in the feces [66]. Broccoli and red cabbage sprouts reduce hepatic cholesterol levels due to their high GL levels [67].

### 4.5. Antiviral Activity

Viral infections are among the major causes of death globally [68]. In the past, various antiviral agents were developed for use in different viral infections: human immunodeficiency virus (HIV), hepatitis B and C, and influenza [68]. However, as a result of their constant clinical use, there have appeared perilous drug-resistant viral strains [69]. The dose-limiting toxic effects of some antivirals in immunocompromised persons also limit the efforts to find a cure for viruses [68]. Hence, scientists have intensified research on developing antiviral agents from plant bioactive molecules to cope up with these challenges [70]. Short-term ingestion of broccoli has been recommended to enhance response to influenza virus-induced markers of inflammation, and also to reduce the virus quantity in predisposed individuals [71]. Besides, consumption of sprouts such as those from mung beans has been reported to reduce viral infection [70]. The relative efficacy of various sprouts or sprout extracts on viruses offers a possibility for research on the future of antiviral phytoagents. The discovery of safe and effective antiviral agents from these extracts may secure humanity against drug-resistant viruses.

### 4.6. Antiatherosclerosis Activity

Cardiovascular diseases remain the chief cause of death in many countries, and atherosclerosis has been categorized as the major condition accounting for the majority of deaths in the United States and Western Europe [72]. The dietary approach to attenuate cardiovascular risk factors is key in the management of atherosclerosis. It has been reported that sprouted seeds are important in the prevention of atherosclerosis. Alteration of steroid excretion by diet modification is a primary means of reducing susceptibility to atherosclerosis. Compounds in broccoli sprouts have been shown to boost the body’s ability to mop up predisposing factors to this condition [73]. In alfalfa, cholesterol–saponin interactions is suggested as the mechanism for antiatherosclerosis activity of sprouts in the in vivo animal model [74]. Thus, alfalfa sprouts are a good dietary source of antiatherosclerosis phytochemicals. Moreover, other biological functions such as antistress activity of sprouts have been described in the literature [75,76]. Thus, consuming sprouts with improved phytochemicals may help reduce effects of stress. Table 1 summarizes bioactive components of alfalfa, buckwheat, broccoli, and red cabbage sprouts along with their health benefits.

## 5. Recent Novel Approach for Enhancing Biological Activities of Sprouts

Recent investigations have shown that sprouting can enhance the accumulation of secondary metabolites. Several techniques have been employed to enhance accumulation of bioactive compounds in germinated seeds. Figure 1 provides a summary of some of the techniques that have been used to improve the bioactive content of sprouted seeds. 

### 5.1. Application of Slightly Acidified Electrolyte Water as an Elicitor in Sprouts

Slightly acidified electrolyte water (SAEW) with a near-neutral pH and chlorine concentration (ACC) can be generated by electrolyzing diluted HCl using a non-membrane electrolytic cell [84]. The SAEW is considered a novel, effective, and relatively inexpensive disinfectant in the food industry [85]. It is environmentally friendly and is regarded as a safe (GRAS) sanitizer. It has been authorized for use in foodstuffs by the Japanese government [86]. In addition to suppressing microbial growth, the SAEW may also influence morphological characteristics and biochemical composition of sprouts [33,86,87]. The SAEW promotes the growth of mung bean sprouts [84,88] since its peroxide (H_2_O_2_) can act as a signal molecule during the germination process [89]. The SAEW could accumulate bioactive compounds in germinated seeds and subsequently improve the sprout bioactivity. It promotes the inhibitory activity of angiotensin I-converting enzyme (ACE) in fermented soybeans [90] alpha glucosidase activity of buckwheat sprouts [91]. SAEW improves the expression of some genes such as Bo-Elong, BCAT, and CY which participate in the synthesis of GLs thereby influencing the content in the germinating seeds [85]. Besides, SAEW promotes the dual accumulation of GABA and rutin of Tartary buckwheat sprouts [33]. The GABA accumulation is the result of SAEW activation of glutamic acid decarboxylase (GAD) [26]. SAEW also possesses regulatory activity for key enzymes such as phenylalanine ammonia-lyase (PAL) involved in phytochemical synthesis during the germination of seeds [33].

### 5.2. Sucrose Treatment

Scientists have extensively investigated the effects of some sugars on the accumulation of phytonutrients in various sprouts [92]. In the process of germination, soluble sugars can accumulate due to the breakdown of stored carbohydrates [92]. Recently, there has been increasing evidence that sucrose treatment of seeds can elevate the levels of various secondary metabolites during germination [32,92]. A 3% sucrose treatment significantly improved antioxidant compounds such as flavonoids, γ-aminobutyric acid, vitamin C, and vitamin E in buckwheat sprouts [93]. Similar observations were made in other studies [78,79]. Wei et al. reported increased vitamin C, total phenolics, and antioxidant activity of mung bean sprouts in the presence of a sucrose solution while Guo et al. described the accumulation of anthocyanins and GLs in broccoli sprouts using sucrose [78,79]. During the germination process of edible seeds, the energy cycle plays a critical role in plant growth and development. It has been shown that exogenous energy increases the levels of bioactive compounds while energy deficit decreases their metabolism in sprouts [94]. Thus, sucrose treatment might act as an external energy source to the seedling which may induce the metabolism of secondary metabolites’ defense responses [95]. The defense-related genes including chalcone synthase and other related proteins are inducible by soluble sugars explaining the role of sucrose in enhancing secondary metabolites in sprouts [95]. In GABA accumulation, sucrose plays a role in the activation of glutamic acid decarboxylase (GAD) which eventually elevates the GABA content of sprouts [96,97]. However, sucrose was also found to suppress the growth of some seeds [92], a phenomenon which may be due to oxidative stress and water deficiency caused by sucrose in seedlings [95].

### 5.3. Irradiation in Sprout Production

Light is one of the indispensable environmental dynamics known to regulate growth and development in plants. Light from various sources, especially sunlight, not only acts as the energy source for the photosynthetic process but also plays a key role in the growth and development throughout the life cycle of a plant [98]. The impact of and response of seedlings to light can be determined by wavelength as well as the quality of light from various sources. During germination, seedlings can perceive a broad spectrum of the wavelengths of light [98]. Therefore, irradiation has become one of the known methods for enhancing the bioactive compounds in sprouting seeds [97,98]. However, the effect of radiation in seedling cellular processes and metabolite biosynthesis can be positive or negative depending on the dose and duration of application [99]. Lim et al. studied how ultraviolet B (UVB) mediates isoflavone accumulation and oxidative–antioxidant system responses in germinating soybeans and observed that the isoflavone content decreased at the UVB of 2 W m^−2^ while the highest increase was observed at the intensity of 1 and 2 W m^−2^ [100]. Therefore, to use irradiation as a bioactive compound elicitor, sprout developers should understand the principle of seed and seedling responses to irradiation. Radiation is also known to improve enzymatic activities such as PAL involved in the synthesis of phytochemicals during the de novo process [100]. Light radiation led to a higher expression of phenylalanine ammonia-lyase (PAL) in germinated flaxseeds significantly improving phenolic acid levels [101] and a higher antioxidant capacity in sweetcorn [102]. The use of artificial light in seed germination is the most recent development in sprout production. The use of blue light-emitting diode (LED) illumination has emerged as an effective technology in sprout production [103].

### 5.4. Computer-Based Prediction Approaches

The temperature of germinating seeds and the duration of sprouting exert a critical influence on the composition of sprouts. Essentially, from the phytochemicals viewpoint, the role of germination time and temperature in sprout bioactivity is related to the effects on enzymatic activity. Time and temperature have a significant influence on the PAL activity, subsequently affecting the synthesis of bioactive compounds in sprouted seeds [104]. Recently, sprout developers have ventured in improving the quality of their products by optimizing sprouting conditions, temperature, and time. Computer-based approaches involving the use of response surface methodology (RSM) allow manipulation and optimization of processes involving independent variables with a combined effect on a given target response [105,106,107]. Today, sprout developers use an RSM plot to visualize the combined effect of the independent variables of time and temperature on the bioactivity of sprouts during germination processes [105,107,108]. The design and the process of using RSM to optimize germination temperature and time for sprouts was described by Paucar-Menacho et al. [109]. The method can be used to project how the accumulation of bioactive compounds in sprouts relates to temperature and time of germination, thereby allowing the manipulation of these variables to have a more optimized set of conditions for the target activity.

### 5.5. Seed Priming

Seed priming is a technique that has been shown to potentially promote rapid and uniform seed germination. The priming process controls the temperature and moisture content of seeds, thereby facilitating seed growth [110]. The technique involves controlled hydration during the early stages of germination to accelerate uniform sprouting [111], and it involves taking seeds through initial phases in the germination process [110]. Commonly, during germination, in phase I, also called the imbibition stage, under appropriate conditions of temperature and moisture, seeds absorb water [110]. In phase II, biochemical changes begin to occur which later trigger seed germination in phase III as seed radicle and shoot emerge. However, in the seed priming, the seed is taken through phase II where the moisture is expelled by drying just before the root can appear [110]. The conditions (temperature and moisture) are optimized to appropriate levels, and the seed is allowed to continue germinating into phase III. Recently, different priming techniques have been investigated and proved to be effective for improving the expression of secondary metabolites in edible sprouts [110,111,112]. The process is known to add nutritional value to sprouts by enhancing the accumulation of phenolic compounds [113,114]. The beneficial effects of seed priming have also been reported in wheat [115], maize [116], broccoli [110], and tomato sprouts [116]. In broccoli sprouts, Hassini et al. reported that a combination of an external elicitor and seed priming under saline conditions promoted growth and GL metabolism in broccoli [111]. Seed priming mechanisms involve epigenetic transformations in the seeds alongside increased concentrations of transcription factors and signaling of proteins when seeds experience water stress [111].

### 5.6. Sprout Biofortification

Biofortification is a process that involves increasing the bioavailability and the concentration of different nutrients in foods. Recently, scientists have discovered the possibility of using biofortification to improve the nutritional quality of edible seeds [117]. For instance, some studies reveal the possibility of increasing iron content by 1.1–15.6 times of sprouted rice grains by first soaking the kernels in a solution of FeSO_4_, before germination [118]. The increase in the iron content may be as a result of the penetration of iron ions through the dorsal vascular bundle present in the endosperm of rice seeds during soaking [118]. A similar result was described for broccoli and radish sprouts obtained by soaking in Fe (III)–EDTA and Fe (III)–citrate solutions before sprouting [119]. Accumulation of selenium by this technique was also reported [120,121]. According to this report, selenium biofortification during sprouting is a recommendable strategy to improve Se minerals in sprouts [121]. Moreover, there is a possibility to obtain Se-enriched wheat kernels by soaking in a 35 mg/L Na_2_SeO_3_ solution [122]. In 4-day-old sprouts of brown rice, the total Se, as well as bound protein, rose to about 60 μmol/L Na_2_SeO_3_ when an external selenite was added. When fully incorporated into the food chain, the biofortification of germinated edible seeds can be a promising vehicle for enhancing the quality of edible sprouts.

### 5.7. Hurdle Approach in Enhancing Functional Properties of Sprouts

Beyond disinfection application, hurdle technology has been used in various food treatments, including improving the functionality of foods [123]. Moreira-Rodríguez et al. described a synergistic effect of ultraviolet A (UVA) and methyl jasmonate on the GLs content of broccoli sprouts [124]. The result showed that the combination led to a higher accumulation of the total GL content in broccoli seeds compared to the individual treatment. Supapvanich et al. studied the combinative effect of salicylic acid and chitosan treatments on the phytochemicals of daikon sprouts during storage [125]. The study indicated that salicylic acid combined with chitosan-treated sprouts enhanced the phenolic compounds and antioxidant activity higher than individual treatments. Higher GL, anthocyanin, phenolic compounds, and vitamin C content were also observed when Chinese kale sprouts were treated with glucose and gibberellic acid [126]. Hence, the hurdle approach can be an effective mechanism to improve the functional property of seeds.

## 6. Microbial Safety of Sprouts

### 6.1. Intervention Strategies for the Microbial Safety of Sprouts

During the germination of seeds, carbohydrates, lipids, and proteins are broken down into simple molecules that can be easily digested and eventually absorbed [12]. Before sprouting, seeds are soaked in a solution and then maintained in a humid environment favorable for sprouting. Bacteria and associated biofilms grow well under conditions with enough moisture and sufficient nutrients. Thus, the settings of sprouting are ideal for bacterial proliferation. The sprouting stage has been categorized as the major source of bacterial contamination in sprouts because bacteria present in the seeds can become internalized in the process of sprouting if not inactivated [127,128]. However, sprouts can be contaminated at any time within the food chain. Washing using clean water and rinsing can minimize infection. Unfortunately, the use of simple methods is not reliable since they cannot remove all the potential pathogens from food surfaces. Furthermore, since sprouts are consumed with minimum processing, high risks are involved in sprout consumption. Sprout-related hazards call for attention and continuous assessment of decontamination methods that will ensure elimination of the pathogenic bacteria and ensure the safety of sprouts [129,130]. Outbreaks involving sprouts have been reported in the past (Table 2). Recently, the United States have recorded an increased number of cases associated with the consumption of sprouts. Between 2010 and 2017, there were major occurrences of multistate outbreaks of *E. coli* linked to sprouts [130]. Clover and alfalfa sprouts were implicated in most of these outbreaks. Between 2011 and 2012, *E. coli* O26 infection associated with sprouts occurred, with the clover sprouts prepared from contaminated seeds reported as the major culprit [130,131]. Physical treatments and chlorine-based compounds have been proposed to improve the microbial safety of sprouts [126]. Thus, due to inability to completely eradicate pathogens from these products, sprouts have become the major fresh products associated with food infection [132,133,134]. As a result, studies have investigated the methods described below as replacements to conventional approaches in sprout decontamination.

### 6.2. Plasma-Based Treatments

Plasma is the fourth state of matter. Generally, plasma treatments involve a partially ionized gas of active plasma-generated species (ultraviolet photons, charged species, and reactive neutrals) [129]. Currently, plasma treatment is used in foods to improve microbial safety and the germination ability of seeds [135,136]. Butscher et al. reported a reduction of 3.4 logarithmic units for *E. coli* on cress seeds upon plasma treatment [129]. The effectiveness of plasma treatment was also observed by Xiang et al. who reported a reduction by 2.32 log_10_ CFU (aerobic bacteria) and 2.84 log_10_ CFU/g (mold and yeast) when mung bean sprouts were treated with plasma-activated water [137]. Puligundla et al. described a reduction by 1.2–2.1 log CFU/g upon 3-minute treatment of radish seeds with a corona discharge plasma jet [138]. The mechanism of microbial inactivation of plasma-based treatments has been described. Mandal et al. stated that the ability to reduce the number of microorganisms with plasma-based treatments involves a chemical interaction between cell membranes of bacteria and radicals (e.g., O, OH), reactive molecules (O_3_ and NO), or charged particles (electrons, molecular ions) released by plasma materials [137]. The researchers further stated that NO and NO_2_ inactivate microorganisms by destroying microbial constituents (proteins, lipids, and nucleic acids). O_3_ and metastable state oxygen (O_2_*) are the major lethal agents in He/O_2_ mixture gases. The role of ultraviolet photons in the inactivation of microorganisms by plasma technology has also been reported [137]. UV photons directly interact with genetic materials of microorganisms to prevent bacterial DNA replication thereby damaging the cells [137]. Moreover, the OH* radicals formed during plasma generation are capable of degrading the bacterial lipid membrane through an oxidation reaction. In the process, OH* degrades the bacterial cells [139,140].

### 6.3. Electrolyte Water (EW)

Over decades, the use of electrolyte water (EW) in the various application has been investigated. Today, EW is gaining popularity as an alternative to the commonly used conventional chlorine-based sanitizers [141]. Electrolyte water (EW), especially acidic electrolyzed water with a pH of 3.66 and chlorine content of 230 mg, has been described as an effective decontaminant in sprouted seeds [142]. The optimum sprout washing time with EW (pH of 3.66 and chlorine content of 230 mg) was 20 s. This was adequate to completely remove aerobic bacteria and molds from the product. EW also reduced the number of microorganisms on the surface of mung beans [88]. It is reported that EW can reduce the number of microorganisms in sprouts with no adverse effect on their sensory properties [141] and thus can be safely used in the food industry for soaking and germinating seeds. The mechanism of microbial efficacy of EW involves several factors. Oh et al. disclosed that the decontamination of EW is largely influenced by pH, oxidation–reduction potential (ORP), the flow rate of electrolytes, and the temperature of the water [140]. Moreover, the lethality of EW is dependent on active chlorine components Cl_2_ and HOCl present in the water [140]. Other constituents of EW such as reactive oxygen species (ROS) can also contribute to the killing of microbes and associated biofilms. HOCl is the most reactive component of EW but its mechanism of microbial decontamination has not been fully understood.

### 6.4. Ultrasound Treatment

Ultrasound treatment is a form of energy produced by sound waves at frequencies above 16 kHz [142]. The ultrasound decontamination method offers advantages due to its low cost, better processing time, improved quality, and reduced damages caused by chemicals and environmental sustainability [143]. Neto et al. evaluated the impact of ultrasound sanitization on microbial and sensory quality of fresh products. The investigation demonstrated a considerable decontamination efficacy with minimal quality damage of products [144]. However, after the treatment with ultrasound, Neto et al. reported that the sample microbial load increased at temperatures higher than 5 °C, indicating that the antimicrobial efficiency of ultrasound can be influenced by temperature levels. For this reason, ultrasound is commonly used alongside other decontamination technologies to maintain an optimal balance between antimicrobial efficacy and sensory damage. The use of ultrasound in combination with other technologies in sprout production has been reported [145]. The mode of decontamination with ultrasound is believed to rely largely on physical effects which lead to microbial cell damage [143]. The role of reactive species such as peroxide hydrogen which disrupts microorganism cells causing cell death has also been described for ultrasound [143]. The report also links the generation of hot spots characterized by localized temperatures (5500 K) and pressures (1000 MPa) with microbial death.

### 6.5. Photosensitization

Information on sprout decontamination using photosensitization is limited. However, the technique is considered to be effective against a variety of microbial pathogens in fresh products. Photosensitization has proved effective in damaging resistant bacteria including bacterial spores and biofilms in products [146]. The decontamination method is effective and is neither mutagenic nor genotoxic, thus can be safely used in disinfecting sprouts [147]. The application of photoactivated chlorophyllin–chitosan significantly inhibited (83%) growth of *Fusarium graminearum* and also delayed fungus growth for 4 days after treatment in wheat sprouts [148]. The decontamination efficacy of the light-based technology is due to the interaction between bacterial cells and the photosensitizer and visible light which releases reactive oxygen species (ROS) and singlet oxygen (1O_2_) [147]. The reactive oxygen species and singlet oxygen ultimately kill microbial cells without damaging the food matrix.

### 6.6. Use of Natural Oil (Essential Oils)

Various kinds of natural oils have been identified in a wide range of herbs and other plant materials. Investigations on the efficacy of essential oils (EOs) in preserving the freshness of sprouts are increasing in number. The ability of thyme oil to kill microbes on seeds and sprouts was reported by Singh et al. who noted at 2.5 mg/L that thyme oil was able to eliminate 6.18 log_10_ CFU/g *E. coli* O157:H7 in alfalfa seeds after 3 min of treatment [149]. The mechanism(s) of action of EOs have not been completely understood. It is reported that EOs are composed of a variety of bioactive compounds and the antimicrobial activity cannot be attributed to only one of these components [150]. Most studies agree on the role played by volatile compounds present in the oils in bacterial decontamination. Components such as phenols disrupts the cell membrane, interfering with their integrity and affecting their functional properties [150]. The volatile substances also cause leakages of the internal cell, depriving the cells of the ability to continue multiplying [149]. The antimicrobial activity of EOs can also be attributed to their hydrophobicity which enables the oils to separate the lipids component of the cell membrane and mitochondria, increasing the permeability of the cells and inhibiting their growth [150]. Other methods involving combined decontamination techniques have been assessed and reported on sprouts [144,150,151]. Further research is still needed to establish the most appropriate combinations that can efficiently reduce the microbial load with minimized overall losses related to the properties of food [152].

### 6.7. Proposed Guides to Reduce the Microbial Hazard in Sprouts

The US Food and Drug Administration outlined various guidelines and recommendations for good agricultural practices for fresh products which may assist sprout developers to prevent risks of contamination during growing, harvesting, transportation, and storage of sprouts [153]. In addition to the US Food and Drug Administration, although not specific to sprouts, the Canadian Food Inspection Agency’s Code of Practice for fresh products also describes frameworks for Good Agricultural Practices needed for minimally processed products which remain relevant to sprout developers [154]. On the other hand, the good manufacturing practice (GMP) system ensures that products are consistently produced according to the outlined quality standards [155]. To produce safe and quality products, the members of the International Sprout Growers Association (ISGA) are required to adhere GMP guidelines [154,155,156,157]. Finally, in today’s climate of consumer interest in food safety matters, sprout developers and regulatory authorities need to often communicate proactively with consumers about hazards related to sprouts and measures to avoid such risks [153]. The benefits of such communication were first recognized in Canada following the 1995 *Salmonella* outbreak and in the USA after an alfalfa sprouts-related microbial infection [154]. The outbreaks were the starting point of media coverage of sprout-related risks. Since then, there has been an immense progress in sprouts safety risk communication.

## 7. New Horizons in Sprout Studies

### 7.1. Could Metabolic Engineering or Biotransformation Be a Solution to the Diversity of Sprout Metabolites?

Plants contain an enormous number of various biologically active chemicals for possible screening. Nevertheless, there is a dilemma in this diversity due to the innate dynamic change of metabolites in plants during seed germination which makes profiling of the desired group of phytochemicals a serious challenge [158]. A possible solution might be provided in using metabolic engineering (ME) or biotransformation. ME is a targeted improvement of cellular properties by either modifying some biochemical pathways or through the introduction of new reactions using the recombinant DNA technology [159,160]. It is recognized as a crucial tool in accumulating natural bioactive compounds in plants [159]. The technique may enable numerous endogenous biochemical pathways to be manipulated during the germination of seeds leading to the generation of targeted secondary metabolites. It has been used to enhance the levels of selected biologically active compounds such as flavonols, quercetin, and kaempferol in plants [159]. ME of bioactive compounds offers an opportunity for further research to allow the production of engineered sprouts. Figure 2 describes the proposed systematic steps to achieve high-yield production of sprout phytochemicals utilizing ME.

On the other hand, biotransformation methods including fermentation and other approaches can be used to improve the quality of sprouts [160]. Germination of seeds combined with biotransformation may produce synergistic effects which may lead to the conversion of various metabolites into new ones for targeted biological functions. A study by Fica et al. indicated that protein supplements from *Spirulina* enhanced the recovery of nutrient-deficient patients; the patients gained weight and their overall health significantly improved [161,162]. So far, spirulina has been used in China as a baby food in baked barley sprouts [161]. Yeast has been successfully used in the production of flavanones through gene expression involving PAL, cinnamate-4-hydroxylase (C4H), 4-coumarate-CoA (4CL), chalcone synthase (CHS) genes [159]. Moreover, after germination, sprouts can be subjected to biotransformation treatments to allow production of desired metabolites as described for buckwheat sprouts fermented using yeast strains [163]. The use of biotransformation not only offers a new approach to enhance the production of the desired plant metabolites, but it also provides possibilities for producing stable products with desirable properties [158].

### 7.2. Green Synthesis of Nanoparticles Using Plant Sprouts

Nanoparticles are useful in the development of sustainable technologies for the future, for both the humanity and the environment. Nanotechnology has generated great enthusiasm in recent years due to its wider application in energy, chemical, and electronic industries. A critical need in this field is to invent a sustainable and ecofriendly process for the synthesis of metallic nanoparticles [164]. The production of nanoparticles by plant products is a green chemistry method that bridges the gap between nanotechnologies and plant biotechnologies [164]. Studies have demonstrated the possibility of using plant sprouts to produce nanoparticles [164]. Nine-day-old alfalfa sprouts and germinated *Brassica juncea* seeds were shown to possess a potential application in the production of silver and Ag–Au–Cu alloy nanoparticles [165]. In another study, alfalfa sprouts showed the ability to act as a natural source for silver nanoparticles [166] while Park and Kim also demonstrated in-vivo synthesis of silver and gold nanoparticles from hydroponically germinated bean, radish, and alfalfa sprouts [166]. Thus, the potential of using plant sprouts in the production of nanoparticles is a promising green technology and is envisaged to present new opportunities in the fight against cancer, neurodegenerative disorders, and many other diseases [165]. However, as Park and Kim argue, this approach possesses critical drawbacks, including potential harmful effects, hence, future studies are needed to help understand the remedies of such shortcomings [167]. Furthermore, despite these interesting discoveries, only limited studies have explored the application of germinated seeds to produce nanoparticles. This and future studies should assess other sprouted seeds to authenticate their ability to successfully synthesize nanoparticles.

### 7.3. The Rise of Sprouting and Gut Health

Gut health is fundamental for human well-being and it helps in the prevention of many chronic diseases influenced by the interactions between gut microbiota and food components [168]. Figure 3 describes the mechanism of interaction between sprout metabolites and gut microbiota. Seed processing approaches such as germination allows the release of metabolites improving their protective capacity in the oxidative process at the cellular level, and also modulating the gut microbiota [168]. All these actions improve gut health and reduce the risk of occurrence of diseases like irritable bowel syndrome, inflammatory bowel disease, obesity, diabetes, colitis, and colorectal cancer [168]. Broccoli sprout juice extracts were found to substantially enhance gut health in a human intestinal cell model [168]. Ferruzza et al. reported that a high content of GLs in broccoli sprouts improved the protection of the intestinal barrier’s integrity in the Caco-2 cells exposed to tumor necrosis factor α under marginal zinc deprivation [168]. In vitro studies have also shown that prebiotic effects in the human colon could be induced by flour extracted from durum wheat sprouts [169]. Moreover, Chen et al. conducted an investigation on the impacts of 7-day-old bean sprouts on the growth and activities of beneficial gut microflora [170]. The results showed that germinated bean seeds significantly promoted the growth and activities of beneficial gut microflora [170].

In a different report, Milán-Noris et al. discovered that germinated chickpeas possessed a great potential to exert anti-inflammatory effects in the lower gut which enhanced the prevention of bowel inflammatory diseases [171]. According to the researchers, sprouted chickpeas are abundant in peptides, proteins, and isoflavones, which have beneficial effects in the gut health [171]. In pardina lentils and green peas, sprouting was reported to reduce gas production by the gut microbiome, thereby reducing flatulence caused when raw seeds are consumed [172]. Therefore, consuming sprouted seeds could improve the gut function and reduce negative outcomes associated with raw seed meals. In general, sprouting is marked by synthesis of bioactive molecules which may possess great benefits to the gut health. An understanding of sprouted seed components behavior during digestion is required to prove the physiological effect of these products on gut health. The interaction between plant phytochemicals and gut microbiota has been described [173]. Therefore, the methods including the in vitro digestion model could be suggested as cheap and useful methods that can offer a quick understanding of the relation between sprouts and gut health. A future study should consider using these approaches to investigate the effects of other sprouted seeds on gut health.

### 7.4. Emerging Uses of Plant Sprouts in Processed Products

The use of sprouted seed flours offers a good strategy to increase the nutritional value and sensory properties of certain food and non-food products. Sprouted seeds can be used to modify the sensory perception of foods, and their addition often differentiate a product within the saturated market space. Sensory parameters such as aroma, appearance, texture, and flavor of products can be distinguished by adding sprouted grains in foods [174]. Due to the increasing amount of simple sugars in seeds during the germination process, sprouted seed flours have been used to improve the overall sweetness of foods, which reduces bitterness [174]. Moreover, during processing of products blended with sprouts, as these foods are being heat-treated, Maillard browning may also occur substantially due to the presence of simple sugars, thus changing the flavor and visual sensory parameters of the final product [174].

Shelf stability is also another important parameter in foods. Addition of sprouted grains can significantly enhance shelf life of foods such as tortillas [174]. This may be due to the high amount of organic acids synthesized in sprouts through germination. Thus, sprouted seeds are promising new sources of ingredients for the fortification of food and non-food materials to improve functionality and acceptability. However, the problem in the use of sprouted flours in products is the excessive α-amylase activity formed during sprouting [175]. Most products with high starch which can gelatinize during baking are susceptible to amylase attack unless the enzyme is inactivated. Despite the tremendous efforts in exploring the possible applications of sprouted seeds, only few products have been successfully produced from the germinated seeds. Thus, this and the future studies should be directed towards assessing more ways to use sprouts in the food chain. Table 3 explored the processed products made by adding germinated grains sprouts.

## 8. Conclusions

Plant sprouts have been widely investigated, and their relationship to human health has been established, including their roles in reducing risks related to chronic conditions (obesity, diabetes, and cytotoxicity), oxidative stress, and microbial safety. In the past decades, numerous advances have been made on sprout development, their biological activities, and applications. However, there still exist areas that have remained partially exploited. For instance, the relationship of sprouts and gut health, application of sprouts in the synthesis of nanoparticles, and the prospect of wider application of sprouted seed flours in processed products are themes that require further research. Germination can be a promising strategy to manipulate plant chemical components and improve their health benefits.

## Figures and Tables

**Figure 1 nutrients-13-02882-f001:**
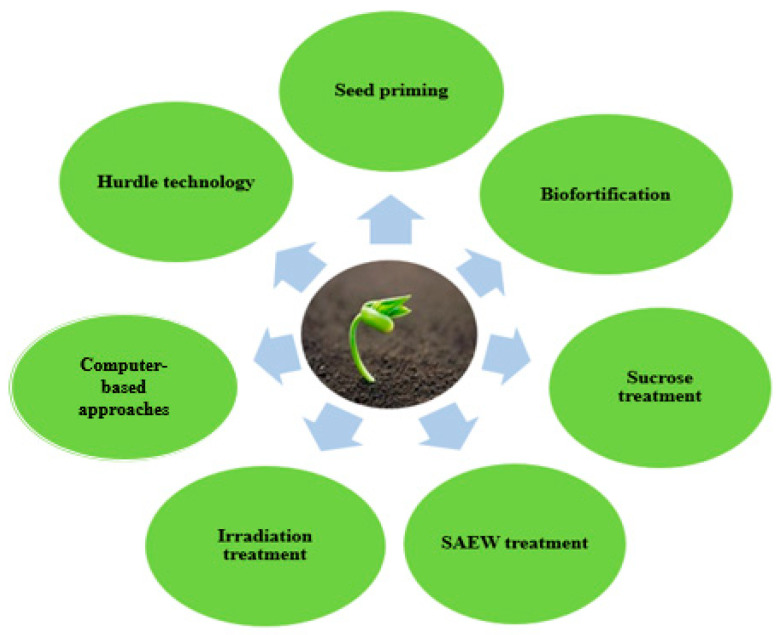
Recent techniques for improving biological activities of edible sprouts.

**Figure 2 nutrients-13-02882-f002:**
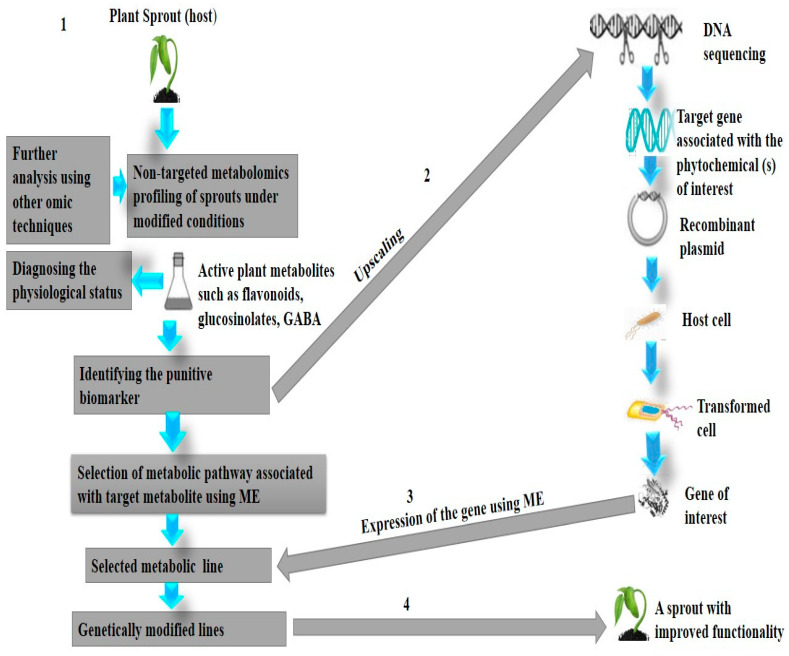
Proposed systematic steps to achieve high-yield production of sprout phytochemicals utilizing M.E. (**1**) Identifying target metabolite(s) in the sprout using various techniques. (**2**) The protein associated with the selected biomarker is identified and upscaled using the recombinant DNA technology. (**3**) Specific metabolic pathway(s) associated with the target phytochemical is manipulated by expressing the gene of interest. (**4**) Eventually, M.E results in an engineered sprout with improved activity for the targeted metabolite.

**Figure 3 nutrients-13-02882-f003:**
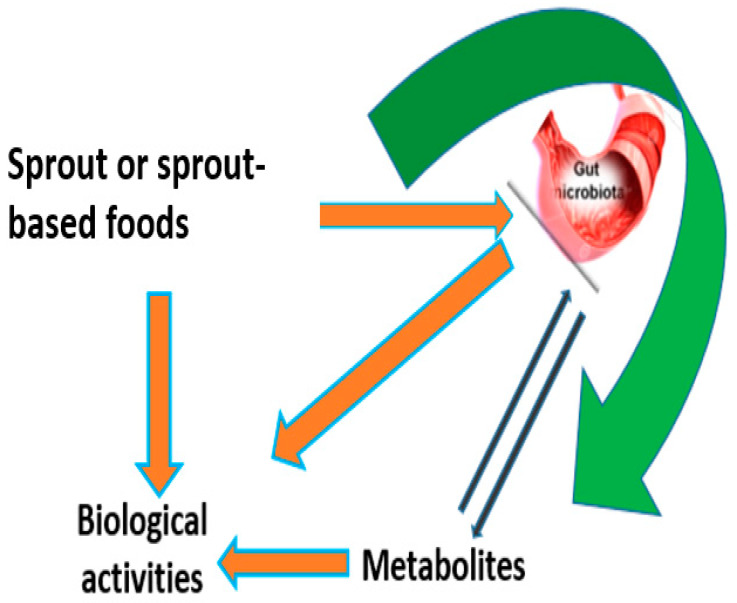
Interaction between secondary metabolites and gut microbiota. When sprout-based foods containing bioactive metabolites are ingested, the microbial ecosystems within the gut rapidly change their functionality in response to the dietary changes [168]. The metabolites are often transformed by the gut microbes before absorption [168]. This transformation modulates their biological activity.

**Table 1 nutrients-13-02882-t001:** Summary of major bioactive compounds in alfalfa, buckwheat, broccoli, and red cabbage sprouts.

Plant Sprouts	Bioactive Compounds	Health Benefits	Reference
Alfalfa	Saponins	Anticancer and antimicrobial activities	[23,24]
	Flavonoids	Anti-inflammatory, antioxidant, antidiabetic activities	[77]
	Phenolic acids (ferulic, garlic, and caffeic acids)	Anti-inflammatory, antioxidant, antidiabetic activities	[77]
	Vitamins C and E and β-carotene	Antioxidant and anti-obesity activities	[20,78,79]
	Trace elements (copper, manganese, selenium)	Antidiabetic and antioxidant activities enhance functions of enzymes	[18,80]
	Coumestrol	Anti-obesity	[81]
Buckwheat	Flavonoids (orientin, vitexin, rutin, and their derivatives) Quercetin, lectins	Anti-inflammatory, hypocholesterolemic, antioxidant, antidiabetic, and anticancer activities	[7,76]
	anthocyanins	Antioxidant and antidiabetic activities	[29]
	2″-hydroxynicotianamine	Antihypertension	[7]
	Aminobutyric acid	Antistress and antioxidant activities	[26,34]
Red cabbage and broccoli	Organic acids (ascorbic acid, aconitic acid, shikimic acid, citric acid, oxalic acid, etc.)	Antibacterial, antioxidant activities	[82,83]
	Glucosinolates (4-methylsulfinylbutyl isothiocyanate)	Anticancer, anti-AGE, hypocholesterolemic, anti-obesity activities	[20,40,45,79]
	Gallic, chlorogenic, sinapinic, benzoic, and ferulic acids, kaempferol	Anti-inflammatory, hypocholesterolemic, antioxidant activities	[39]
	Anthocyanin	Anticancer, antioxidant, anti-inflammatory activities	[35,36]

**Table 2 nutrients-13-02882-t002:** Summary of microbial infection outbreaks associated with sprouts from 2010 to 2020.

Bacteria	Year	Prevalence	Source (Sprouts)	Country	Reference
*E. coli* O103	2020	51	Clover sprouts	USA	https://www.cdc.gov/ecoli/2020/o103h2-02-20/index.html. Accessed date (12 April 2021)
*Salmonella*	2016	26	Alfalfa	USA	[132]
*E. coli O121*	2014	19	Alfalfa	USA	[130]
*E. coli O26*	2012	29	Raw clover	USA	[131]
*E. coli* *O104:H4*	2011	3842	Fenugreek	Germany	[127]
*Salmonella*	2010	190	Bean sprouts	UK	[127]

**Table 3 nutrients-13-02882-t003:** Examples of studies reporting on the addition of sprouts in processed products.

Sprout	Product Added	Function	Reference
Wheat	Breadmaking	Modify pasting characteristics	[176]
Soybean	Cosmetics	Whitening agent	[177]
Wheat	Tortillas	Enhance shelf life and sensory attributes	[174]
Brown rice	bread	Improve textural properties	[178]
Sorghum	Bread	Soften the dough	[179]
Brown rice	noodels	Improves quality properties	[180]
Maize	Cookies	Modify gelatinization properties	[181]
Quinoa and oat	Bread	Improve the nutritional value	[182]
Wheat	Yoghurt	Improve a broad range of quality characteristics	[183]
Barley	Beer brewing	Improve the beer’s flavor, taste, and nutritional value	[184]
Oat	Beer brewing	Improve the aroma	[185]
Soybean, brown rice	Steamed buns	Enhance overall nutritional quality	[184]
Brown rice	Wine	Improve organoleptic properties	[184]
Barley, rice	Vinegar	Enhance the enzymatic activity and aroma	[184]
Soybean	Soy sauce	Improve organoleptic properties	[186]
Millet, soybean	Biscuits	Improve quality parameters (hardness, stickiness)	[187]
Pigeon pea	Semolina pasta	Increase the nutritional value	[188]
Alfalfa and flax	Hen egg	Enrichment lowering the cholesterol	[189]
Buckwheat	Pastor	Improve functionality	[190]
Millet	Milk beverage	Enhance the nutritional value	[191]

## Data Availability

Not applicable.

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
