# Peer review of "Edible Plant Sprouts: Health Benefits, Trends, and Opportunities for Novel Exploration"

_nutrients, 2021, doi:10.3390/nu13082882_

Round 1

Reviewer 1 Report

The present manuscript is well organized and interesting to read, however I see the following minor issues that should be resolved:

In the abstract:  "However, there are still numerous issues that remain partially addressed" - please exemplify 2-3 examples, same in the manuscript.

In the conclusion section: " However, there still exist areas that have remained partially exploited." - please exemplify some of them.

Line 726-727: Plant sprouts have been widely investigated, and their relationship to human health established to some extent - please exemplify to which extend...

Lines 736-737: "The apprach is also critical in support designing
recommendations on extensive use of plant sprouts in promoting health."
- what do you mean?

Figure 4. Interaction between secondary metabolites and gut microbiota - i could not see any link or explanation related to this figure!?

Literature is up to date.

Author Response

Dear Editor,

RESPONSE TO REVIEWER COMMENTS

We are grateful for your valuable comments and we have carefully revised the manuscript as the reviewers suggested. Please find the response to the reviewer comments. Comments are in red and responses in black.

Regards,

Deog-Hwan Oh (PhD)

Reviewer #1

Comment: In the abstract:  "However, there are still numerous issues that remain partially addressed" - please exemplify 2-3 examples, same in the manuscript.

Response: The lines have been modified as follows: “However, subjects such as application of transgenic technologies in sprout production, utilizing sprouts as leads in drug discovery, assessing the dynamics of how the sprout phytochemical interact with drugs are areas that require special attention for future clinical exploration” (Line 20-23)

Comment: In the conclusion section: “However, there still exist areas that have remained partially exploited." - Please exemplify some of them.

Response: the conclusion has been modified as follows: “However, there still exist areas that have remained partially exploited." - Please exemplify some of them. For instance, the dynamic of interaction between sprout biologically active compounds and biochemical pathways in the body, the interactions with drugs, and the prospect of using sprout active compounds as leads in drug development are themes that require further future research. These areas offer avenues for further exploration” (Line 699-703).

Comment: Line 726-727: Plant sprouts have been widely investigated, and their relationship to human health established to some extent - please exemplify to which extend.

Response: this line has been modified as follows:  Plant sprouts have been widely investigated, and their relationship to human health have been established including their roles in reducing risks related to chronic conditions (obesity, diabetes, and cytotoxicity), oxidative stress, and, and microbial safety (Line 706-710)

Comment: Lines 736-737: "The approach is also critical in support designing
recommendations on extensive use of plant sprouts in promoting health." - What do you mean?

Response: this line has been deleted

Comment: Figure 4. Interaction between secondary metabolites and gut microbiota - I could not see any link or explanation related to this figure!?

Response: The description of this figure has been inserted as foot note as follows:

“Interaction between secondary metabolites and gut micro biota. When sprout based foods containing bioactive metabolites are ingested, the microbial ecosystems within the gut rapidly change their functionality in response to the dietary changes. The metabolites are often transformed by the gut microbes before absorption. This transformation modulates their biological activity” (Line 693-698)

Reviewer 2 Report

The presented review is quite interesting, but should be substantially rewritten. My general impression is that the Authors started a few interesting issues but none of them is deeply studied. I would suggest to focus on the papers that consider sprouts themselves, excluding all the information on the activity of the single compounds, that can be found in the sprouts. I would also suggest to expand the part about the role of different sprouts on preventing some diseases (e.g. gastrointestinal tract, diabetes, athreosclerosis, cancer) to make the review more profound. I would also recommend to delete paragraphs 6 and 7, as the information presented there  (especially paragraph 6) may be a subject of another review. Paragraph 7 might be misleading to the readers, as it considers single compounds.

Some minor suggestions are as follows:

  • line 125: should be tartary buckwheat, not tertiary
  • paragraph 4.2: anticancer is a term used for animal studies, with induced tumors. I would suggest rather "cytotoxic activity"
  • paragraph 4.3, 4.4, 4.5: please delete the information concerning the activity of single compounds, and leave only those on the sprouts extracts
  • paragraph 4.6: I do not undestand the "anti-stress" activity, I suggest to delete the paragraph
  • line 391: avrious means various?
  • Figure 2: low quality, should be improved
  •  

Author Response

Dear Editor,

RESPONSE TO REVIEWER COMMENTS

We are grateful for your valuable comments and we have carefully revised the manuscript as the reviewers suggested. Please find the response to the reviewer comments. Comments are in red and responses in black.

Regards,

Deog-Hwan Oh (PhD)

Reviewer #2

Comment: I would also suggest to expand the part about the role of different sprouts on preventing some diseases (e.g. gastrointestinal tract, diabetes, atherosclerosis, and cancer) to make the review more profound.

Response: diabetes and cancer activities of sprouts has already been discussed in the review. However, to the best of our knowledge, no report has been published related to gastrointestinal tract activities of sprouts.

The following paragraph has been added for atherosclerosis according the above reviewer’s recommendation.

“Cardiovascular diseases remain chief cause of death in many countries, and atherosclerosis, has been categorized as the major condition accounting for the majority of deaths in United States and Western Europe [72]. Dietary approach to attenuate cardio-vascular System is key in management of atherosclerosis. It has been reported that sprouted seeds are important in prevention of atherosclerosis. Alteration of steroid excretion by diet modification is a primary means of reducing susceptibility to atherosclerosis. Compounds in broccoli sprouts have been shown to boost the body's ability to mop up predisposing factors to this condition [73]. In alfalfa, cholesterol-saponin interactions is suggested as the mechanism for anti-atherosclerosis activity of the sprouts in the in vivo animal model [74]. Thus, alfalfa sprouts are good dietary source of ant-atherosclerosis phytochemicals” (Line 220-231)

Comment: I would also recommend to delete paragraphs 6 and 7, as the information presented there (especially paragraph 6) may be a subject of another review. Paragraph 7 might be misleading to the readers, as it considers single compounds.

Response: Paragraph 6 and 7 were modified accordingly. From paragraph 6, the following information was deleted “The γ-aminobutyric acid (GABA) accumulation in the buckwheat sprouts was reported and is related to the activity of the glutamic acid decarboxylase enzyme [33]. GABA helps in lowering blood pressure and inhibiting the proliferation of cancer cells in the body [34].  Recently, 2″-hydroxynicotianamine (2HN) has also been described as a functional ingredient in buckwheat seeds and sprouts [7].”

From paragraph 7, the following information was deleted

 “About 120 different GLs have been identified in broccoli [43]. Among these, glucoraphanin is the dominant and most known GLs in broccoli sprouts composing 81% of the total GLs content [44]. The GLs in the vegetables are inactive, but they can be hydrolyzed to generate the active form, sulforaphane (4-methylsulfinylbutyl isothiocyanate) in plants and upon digestion in humans [45]. Myrosinase enzyme, a family of enzymes involved in plant defense mechanism and also present in the human gut catalyzes the conversion of GLs into active form improving”

However, we would like to clarify that the paragraph 6 and 7 alongside the previous paragraphs (4 and 5) were only focused on the phytochemicals that have been reported in the respective plant sprouts. The activities of these sprouts are covered independently in the subsequent paragraphs. Consequently, we opted to modify the paragraphs to make them clearer instead of deleting them.

Some minor suggestions are as follows

Comment: line 125: should be tartary buckwheat, not tertiary

Response: the line 125 has been corrected and the “tertiary” was changed to “tartary” (line 129)

Comment: paragraph 4.2: anticancer is a term used for animal studies, with induced tumors. I would suggest rather "cytotoxic activity"

Response: in paragraph 4.2, the term ‘anticancer’ was replaced by ‘cytotoxic activity”

Comment: paragraph 4.3, 4.4, 4.5: please delete the information concerning the activity of single compounds, and leave only those on the sprouts extracts

Response

From paragraph, 4.3 the following information were deleted

‘The sulforophane (SFN) in broccoli and red vegetable sprouts is regarded as an anti-diabetic biomarker in these sprouts’

‘At a concentration of 500μg, SFN extract exhibited a strong inhibition of amylase enzyme indicating the potent anti-diabetic function of broccoli and red cabbage’

‘Polyphenolic compounds such as rutin contents present in the majority of these sprouts are antidiabetic agents inhibiting elevating blood sugar levels in the body’.

‘The potent inhibitory effect of rutin has been reported both in vitro and in-vivo experiments’

From paragraph 4.4, the following information were deleted

‘Alfalfa sprouts are rich in saponins and may offer a potential pharmacologic utility in the treatment of hypercholesterolemia’

‘On the other hand, the naturally occurring saponins in alfalfa sprouts have been associated with inhibition of cholesterol absorption and reduction in the blood’

From paragraph 4.5, the following information was deleted

‘Flavonoids and their derivatives are among the most known antiviral phenolic compounds present in a variety of plant sprouts’

Comment: paragraph 4.6: I do not understand the "anti-stress" activity, I suggest to delete the paragraph

Response: the paragraph has been deleted accordingly.

Comment: line 391: avrious means various?

Response: the spelling error has been corrected and avrious was changed to “various” accordinglyn (line 361).

Comment: Figure 2: low quality, should be improved

Response: The figure 2 has been modified accordingly as follows (line 560)

Round 2

Reviewer 2 Report

The review was improved but still some of my issues were not taken into account. Generally I suggested to delete paragraph 6 and 7, but some misunderstanding occured, as the paragraphs are still included. I meant paragraph 6 "Microbial safety" and paragraph 7 "Future perspectives". Especially the latter paragraph does not sound scientific to me in its present form: except section 7.1, the other sections concerns single compounds, not the sprouts extracts, and such approach does not make sense in the review on sprouts themselves. Thus, I strongly recommend to delete sections 7.2 - 7.4, and substantially shorten the paragraph on microbial safety and include the information into "Future perspectives" paragraph.

Moreover, the title of paragraph 4.2 should be "Cytotoxic activity", and the first sentence of the paragraph should be corrected, as no "cytotoxic-related" diseases exist.

Author Response

Dear Editor,

RESPONSE TO REVIEWER COMMENTS

We are grateful for your valuable comments and we have carefully revised the manuscript as the reviewers suggested. Please find the response to the reviewer comments. Comments are in red and responses in black.

Regards,

Deog-Hwan Oh (PhD)

Reviewer #2

Comment: Thus, I strongly recommend to delete sections 7.2 - 7.4

Response: sections 7.2 and 7.4 were deleted and replaced by new information as “Green synthesis of nanoparticles using plant sprout” and “Emerging uses of plant sprouts in processed products,” respectively (7.2-line 557; 7.4-line 619). The previous table 3 was also replaced by new one titled “Examples of studies reporting on the addition of sprouts in processed products”

Since the reviewer strongly recommend that the paragraph 7 should be deleted because it contained individual compounds instead of sprouts themselves, we opted to also delete the old form of section 7.3 and replace it with a new information with heading “The rise sprouting and gut health” (line 577).

Comment: substantially shorten the paragraph on microbial safety and include the information into "Future perspectives" paragraph.

Response: this paragraph was substantially shortened and some information deleted as follows

The previous section “6.2.6. Hurdle Methods” was deleted

Previous sub sections “6.2.7.2. Application of Hazard Analysis Critical Control Points (HACCP) and Good Manufacturing Practices (GMPs)” was deleted.

The previous subsection “6.2.7.3. Sprouts Safety Risk Communication” was deleted.

However, we wish that this section remains as it not to be combined with future perspective section since we treated it independently. In this paragraph, “6” our aim was to report on the previous microbial related cases on sprouts and also investigate on the various decontamination techniques that have been used on sprouts. Therefore, we wish that if need be, with your recommendation, we can modify the information within this section instead of deleting it.”

Comment: Moreover, the title of paragraph 4.2 should be "Cytotoxic activity", and the first sentence of the paragraph should be corrected, as no "cytotoxic-related" diseases exist.

Response: ‘The “Cytotoxic activity” title was inserted for this section, and the "cytotoxic-related" corrected (Line 155 and 156
